# Optimal Brain Compression: A Framework for Accurate Post-Training Quantization and Pruning

**Elias Frantar** *
IST Austria
elias.frantar@ist.ac.at

**Sidak Pal Singh**
ETH Zurich
sidak.singh@inf.ethz.ch

**Dan Alistarh**
IST Austria & Neural Magic
dan.alistarh@ist.ac.at

## Abstract

We consider the problem of model compression for deep neural networks (DNNs) in the challenging one-shot/post-training setting, in which we are given an accurate trained model, and must compress it without any retraining, based only on a small amount of calibration input data. This problem has become popular in view of the emerging software and hardware support for executing models compressed via pruning and/or quantization with speedup, and well-performing solutions have been proposed independently for both compression approaches. In this paper, we introduce a new compression framework which covers both weight pruning and quantization in a unified setting, is time- and space-efficient, and considerably improves upon the practical performance of existing post-training methods. At the technical level, our approach is based on an exact and efficient realization of the classical Optimal Brain Surgeon (OBS) framework of [LeCun, Denker, and Solla, 1990] extended to also cover weight quantization at the scale of modern DNNs. From the practical perspective, our experimental results show that it can improve significantly upon the compression-accuracy trade-offs of existing post-training methods, and that it can enable the accurate *compound* application of both pruning and quantization in a post-training setting.

## 1 Introduction

The impressive recent progress of deep learning for solving challenging tasks across several domains has been accompanied by a significant increase in parameter counts and computational costs for executing such models. A natural consequence has been a growing effort to reduce such costs via *model compression*, and the two most popular approaches for model compression are *pruning*—removing neural network weights by setting them to zero—and *quantization*, reducing the precision at which neural network weights and activations are stored and manipulated. Hundreds of such pruning and quantization approaches have been proposed and analyzed [17, 11], with the general goal of obtaining efficient variants of deep neural nets (DNNs) which would preserve accuracy while maximizing compression. Despite impressive progress, compression is still a laborious process: the pruning and quantization stages are often done independently, and recovering model accuracy after compression often requires partial or even full retraining of the compressed model.

An alternative but challenging scenario is the *post-training compression* setup [31, 24, 19, 25], in which we are given a trained but uncompressed model, together with a small amount of *calibration data*, and must produce an accurate compressed model in *one shot*, i.e. a single compression step, without retraining, and with limited computational costs. This is motivated by practical scenarios such as the MLPerf Inference Benchmark [37], and is the setting we focus on in this paper.

---

*Corresponding author.

36th Conference on Neural Information Processing Systems (NeurIPS 2022).

Compression via *weight pruning* started with seminal work by LeCun et al. [23], complemented by Hassibi and Stork [13], who proposed a mathematical framework called the *Optimal Brain Surgeon (OBS)*, for choosing the set of weights to remove from a trained neural network, by leveraging second-order information. (We describe their approach in Section 3.) Recent advances, e.g. [6, 40, 39, 10] showed that OBS can lead to state-of-the-art compression at DNN scale, by introducing numerical methods which can approximate the second-order information needed by OBS at the massive parameter counts of modern models. However, these approaches do not apply to the *post-training* setting, as they require gradual pruning, as well as significant retraining, to recover good accuracy.

An alternative approach, which is standard in the context of *post-training* compression, has been to break the compression task into layer-wise sub-problems, identifying a compressed weight approximation for each layer, given a sub-sample of the layer's inputs and outputs based on calibration data. This line of work, e.g. [42, 31, 19], introduced elegant solvers for the resulting layer-wise weight quantization problem, which achieve state-of-the-art results for post-training quantization. Recently, AdaPrune [18] showed that this approach can also be effective for post-training weight pruning.

In this context, a natural question is whether existing approaches for pruning and quantization can be *unified* in order to cover both types of compression in the post-training setting, thus making DNN compression simpler and, hopefully, more accurate. This question is also of practical importance, since both GPU and CPU platforms now *jointly* support sparse and quantized formats [30, 35], and, as we illustrate experimentally, the resulting models could be executed with compound speedups.

**Contribution.** In this paper, we provide a mathematical framework for compression via pruning or quantization, which leads to state-of-the-art accuracy-versus-compression trade-offs in the challenging *post-training compression* setup. Our framework starts from the layer-wise compression problem described above, by which the global compression task, defined either for pruning or quantization, is first split into layer-wise sub-problems, based on the layer behavior on the calibration data. Specifically, given a layer $\ell$ defined by weights $\mathbf{W}_\ell$, and layer inputs $\mathbf{X}_\ell$, we aim to find a compressed version of the weights $\widehat{\mathbf{W}}_\ell$ which minimizes the output difference relative to the uncompressed layer, measured via the squared error between the original and compressed layer, acting on the sample input $||\mathbf{W}_\ell\mathbf{X}_\ell - \widehat{\mathbf{W}}_\ell\mathbf{X}_\ell||_2^2$, under a fixed compression constraint on $\widehat{\mathbf{W}}_\ell$.

Although solving this problem optimally for sparsity or quantization constraints is NP-hard [2, 31], it is a key step in all state-of-the-art post-training compression methods, both for pruning [18, 9] and for quantization [31, 18, 24]. Once this is solved per layer, a solution to the global problem can be obtained by combining layer-wise solutions, which is handy especially for non-uniform compression, e.g. [15, 9]. Thus, several approximations for this problem have been proposed [31, 19, 18].

We show that there still is significant room for improvement when solving the layer-wise compression problem. Roughly, our approach is to specialize the OBS framework to the squared error formulation above: in this case, the framework can in theory produce an exact greedy solution, but a direct implementation would have infeasible $\Theta(d^4)$ computational cost, where $d$ is the layer dimension. Our main technical contribution is a series of algorithms which reduce this computational cost, *without any approximations*, to $O(d \cdot d_{col}^2)$ where $d_{col}$ is the column dimension of the weight matrix. In practice, these improvements are significant enough to allow us to implement the exact OBS greedy solution, which prunes *one weight at a time*, and updates *all remaining weights* after each step, at the scale of modern DNNs with tens of millions of parameters, within reasonable time, on a single GPU. We provide efficient implementations of our methods at `https://github.com/IST-DASLab/OBC`.

In turn, this algorithmic development allows us to apply the OBS approach to *quantization*. The resulting algorithm, called the *Optimal Brain Quantizer (OBQ)*, quantizes weights iteratively one-at-a-time, depending on their impact on the loss increase, after which it applies a closed-form update to the remaining unquantized weights, further reducing the loss. This solves the two problems efficiently, and in a unified manner—we call the unified framework the *Optimal Brain Compressor (OBC)*.

**Experimental Results.** We apply OBC to standard tasks and models covering image classification, object detection, and language modelling applications. We first show that our framework yields significantly better solutions for the layer-wise compression problem, which leads to higher-accuracy end-to-end compressed models for both pruning and quantization, relative to the corresponding state-of-the-art techniques, often by significant margins. Second, we show that our pruning and quantization approaches can be compounded, with surprisingly strong results: we obtain a $12\times$ reduction in theoretical operations with a 2% accuracy drop for GPU-supported compound compression [30],

and a $4\times$ speedup in *actual runtime* with only $1\%$ accuracy drop for a CPU-based sparsity-aware runtime [35]. Together, these results suggest for the first time that post-training compression can be competitive with full retraining.

## 2   Related Work

**Optimal Brain Surgeon (OBS).** The classic OBS framework [23, 13] was originally applied to networks with hundreds of weights; more recently, methods such as WoodFisher [39] rendered the approach computationally feasible for DNNs by using a block-diagonal Fisher approximation of the Hessian, while follow-up methods introduced more efficient and general algorithms for handling the inverse Fisher matrix, or customize this approximation to specific model families [21]. Earlier work called Layer-wise OBS (L-OBS) [6] was inspired by the K-FAC approximation [29, 12]: L-OBS approximates the OBS framework not for the global objective, but for a quadratic per-layer loss, while also pruning all weights based on a single Hessian computation. At a high level, our approach is similar, in that we apply OBS layer-wise; however, we apply OBS *exactly*, that is, pruning one weight at a time, and exactly recomputing the Hessian after every pruning step. This is made computationally tractable by several new algorithmic ideas, and yields significantly improved results relative to L-OBS. This prior work on pruning considered settings with extensive finetuning. By contrast, we will focus on the post-training setting, where only a small amount of calibration data is available.

**Post-Training Quantization.** This setting has been primarily considered for quantization, and most state-of-the-art methods work by performing layer-wise compression. Specifically, BitSplit [6] optimizes the quantized weights bit by bit, while AdaRound [31] finds a weight rounding policy through gradient based optimization with an annealed penalty term that encourages weights to move towards points on the quantization grid. AdaQuant [19] relaxes the AdaRound constraint, allowing weights to change during quantization-aware optimization, via straight-through estimation [33]. BRECQ [24] suggested that accuracy can be improved further by integrating second-order information into the layer-wise losses and by jointly optimizing hand-crafted blocks of related layers.

A key step of AdaRound, AdaQuant and BRECQ is to quantize layers incrementally, in *sequential* order, so that errors accumulated in earlier layers can be compensated by weight adjustments in later ones. This significantly improves performance, but reduces flexibility, as the entire process may need to be re-done whenever we wish to change compression parameters of one layer. We instead target *independent* compression of each layer, allowing the end model to be simply "stitched" together from layer-wise results. Despite operating independently on each layer, we find that, after correcting basic statistics such as batchnorm, our method performs on par to sequential ones for uniform quantization.

**Post-Training Sparsification.** The layer-wise approach was shown to also be effective for post-training pruning by AdaPrune [18], which pruned weights to the GPU-supported N:M pattern [45]. AdaPrune first drops parameters according to their magnitude [46] and then reoptimizes the remaining weights to reconstruct the pre-compression calibration set output. This is similar to [16, 8] which also perform layer-wise reoptimization of the remaining weights. Follow-up work [10] noted that the results of AdaPrune can be improved further by performing more frequent pruning/optimization steps. Our algorithm pushes this idea to the limit, performing *full reoptimization* after every single pruned weight, while remaining computationally tractable. We further use a more sophisticated weight selection metric which incorporates second-order information. Finally, [10] also introduces *global AdaPrune*, a more expensive global optimization step applied on top of the layer-wise AdaPrune results, which can bring additional accuracy gains. This can also be applied to our pruned models.

**Non-Uniform Compression.** An orthogonal practical question is how to compress different layers to maximize accuracy under a given resource constraint, such as latency or energy consumption. Existing methods can be roughly categorized into search-based and solver-based approaches. The former, e.g. AMC [15] or HAQ [41], search for a layer-wise compression policy directly via, for example, reinforcement learning or genetic programming [43], whereas the latter, e.g. HAWQv3 [44] or AdaQuant [19], construct a relaxed version of the overall problem that is then solved exactly. We focus here on solver-based approaches, as they can rapidly adapt to different scenarios when combined with accurate independent layer-wise compression schemes; however, our techniques could be of interest for search-based methods as well. Concretely, we use the problem formulation of AdaQuant [19] to which we apply the DP algorithm of SPDY [10] to achieve fast solving times even with a large number of possible choices per layer.

# 3 Problem Definition and Background

**The Layerwise Compression Problem.** Following prior work on post-training compression, e.g. [31, 19], we define the problem as follows. Mathematically, we model a layer $\ell$ as a function $f_\ell(X_\ell, W_\ell)$ acting on inputs $X_\ell$, parametrized by weights $W_\ell$. The goal of layer-wise compression is to find a "compressed" version of $W_\ell$ that performs as similarly as possible to the original weights. More formally, the compressed weights $\widehat{W}_\ell$ should minimize the expected layer output change as measured by some loss $\mathcal{L}$ while at the same time satisfying a generic compression constraint, which we denote by $\mathcal{C}(\widehat{W}_\ell) > C$, which will be customized depending on the compression type:

$$\operatorname{argmin}_{\widehat{W}_\ell} \quad \mathbb{E}_{X_\ell} \mathcal{L}(f_\ell(X_\ell, W_\ell), f_\ell(X_\ell, \widehat{W}_\ell)) \quad \text{subject to} \quad \mathcal{C}(\widehat{W}_\ell) > C. \tag{1}$$

The expectation over the layer inputs $X_\ell$ is usually approximated by taking the mean over a small set of $N$ input samples. This low-data setting is one of the primary applications of layer-wise compression. Further, most works [42, 31, 19] focus on compressing linear and convolutional layers, which can be unfolded into linear ones, as these are prevalent in practice, and use the squared loss to measure the approximation error. This definition of the loss can be motivated, via a sequence of approximations, from second-order information: please see [31] for a precise derivation. Furthermore, this approximation approach has been shown to work well in many applications [31, 19, 9].

We follow these conventions as well, and work with the specific layer-wise compression problem stated formally below, where the weights $\mathbf{W}_\ell$ are a $d_{\text{row}} \times d_{\text{col}}$ matrix (for conv-layers $d_{\text{col}}$ corresponds to the total number of weights in a single filter), and the input $\mathbf{X}_\ell$ has dimensions $d_{\text{col}} \times N$.

$$\operatorname{argmin}_{\widehat{\mathbf{W}}_\ell} \quad ||\mathbf{W}_\ell \mathbf{X}_\ell - \widehat{\mathbf{W}}_\ell \mathbf{X}_\ell||_2^2 \quad \text{s.t.} \quad \mathcal{C}(\widehat{\mathbf{W}}_\ell) > C. \tag{2}$$

**The Optimal Brain Surgeon (OBS) Framework.** The OBS framework [23, 13] considers the problem of accurately pruning a trained dense neural network. It starts from the Taylor approximation at the given point (assumed to have negligible gradient), and provides explicit formulas for the optimal single weight to remove, as well as the optimal update of the remaining weights which would compensate for the removal. More precisely, let $\mathbf{H}$ denote the Hessian matrix of the loss at the given (dense) model. Then the weight to prune $w_p$ which incurs the minimal increase in loss and the corresponding update of the remaining weights $\boldsymbol{\delta_p}$ can be calculated as follows:

$$w_p = \operatorname{argmin}_{w_p} \frac{w_p^2}{[\mathbf{H}^{-1}]_{pp}}, \quad \boldsymbol{\delta_p} = -\frac{w_p}{[\mathbf{H}^{-1}]_{pp}} \cdot \mathbf{H}_{:,p}^{-1}, \tag{3}$$

where $[\mathbf{H}^{-1}]_{pp}$ denotes the $p$th diagonal entry of the inverse Hessian, and $\mathbf{H}_{:,p}^{-1}$ is its $p$th column.

**OBS for Layer-Wise Pruning.** We will now instantiate this framework for the layer-wise pruning problem, defined above. First, the loss in equation (2) is quadratic and since our starting point is given by the dense weights achieving the minimal loss of 0, the assumptions of the OBS framework are fully met, meaning that its formulas are *exact* for this specific problem formulation. Thus, iterating the OBS framework to remove one weight at a time would yield an exact *greedy solution* for the layer-wise pruning problem, as it takes the (locally) optimal decision at each step. While this greedy approach does not guarantee convergence to a global optimum, such approaches can be very effective for dealing with problem instances that are too large to be handled by exact methods.

# 4 An Optimal Greedy Solver for Sparsity

The obvious challenge is that applying the OBS framework in its true form, i.e. pruning a single weight at a time using the exact formulas in (3), is computationally very demanding. The Hessian $\mathbf{H}$ is a $d \times d$ matrix where $d = d_{\text{row}} \cdot d_{\text{col}}$, which is already expensive to store and compute with. Additionally, this matrix needs to be updated and inverted at each of the $O(d)$ steps with a computational complexity of $\Theta(d^3)$. Clearly, an $O(d^4)$ total runtime is too inefficient for pruning most layers of modern neural networks, as $d$ is usually $\geq 10^5$ or even $\geq 10^6$ for several layers. However, as we will now show, it is actually possible to reduce the overall costs of this process to $O(d_{\text{row}} \cdot d_{\text{col}}^3)$ time and $\Theta(d_{\text{col}}^2)$ memory, making it efficient enough to prune e.g. all layers of a medium-sized model such as ResNet50 in a bit more than one hour on a single NVIDIA RTX 3090 GPU. We emphasize that the techniques we introduce are exact; unlike prior work [6, 39], we do not rely on any approximations.

**The ExactOBS Algorithm.** In the following, we introduce our efficient instantiation of the OBS framework, for the layer-wise compression problem, which we call ExactOBS, in step-by-step fashion. We start by rewriting the matrix squared error in (2) as the sum of the squared errors for each row in the weight matrix. As we are always dealing with a fixed layer $\ell$, we drop the subscript $\ell$ to simplify notation. The objective is then equivalent to $\sum_{i=1}^{d_{\text{row}}} ||\mathbf{W_{i,:}X} - \widehat{\mathbf{W}}_{\mathbf{i,:}}\mathbf{X}||_2^2$.

This way of writing the error makes it clear that removing a single weight $[\mathbf{W}]_{ij}$ only affects the error of the corresponding output row $\mathbf{Y_{i,:}} = \mathbf{W_{i,:}X}$. Hence, there is no Hessian interaction between different rows and so it suffices to work only with the individual $d_{\text{col}} \times d_{\text{col}}$ Hessians corresponding to each of the $d_{\text{row}}$ rows. Further, as the dense layer output $\mathbf{Y} = \mathbf{WX}$ is fixed, the objective for each row has standard least squares form and its Hessian is given by $\mathbf{H} = \mathbf{2XX}^\top$.

Although this observation already reduces computational complexity, two key challenges remain: (a) applying OBS to each row still costs $O(d_{\text{col}} \cdot d_{\text{col}}^3)$ time, which is too slow for large layers, and (b) we need fast access to the Hessian inverses of all $d_{\text{row}}$ rows, since we want to prune the minimum score weight of the whole matrix rather than just per row in each step. In particular, (b) requires $O(d_{\text{row}} \cdot d_{\text{col}}^2)$ GPU memory, which is likely to be infeasible.

**Step 1: Handling a Single Row.** We first describe how to efficiently prune weights from a single row with $d_{\text{col}}$ parameters. For simplicity, we denote such a row by $\mathbf{w}$ with corresponding Hessian $\mathbf{H}$. The full algorithm for this procedure is given in Algorithm 1; in the following, we provide a detailed description. The key idea is to avoid having to do the full $\Theta(N \cdot d_{\text{col}}^2)$ calculation and $\Theta(d_{\text{col}}^3)$ inversion of $\mathbf{H}$ in each step. The former is easy, as the weights themselves do not enter the calculation of $\mathbf{H} = \mathbf{2XX}^\top$, and the Hessian for the weights with pruning mask $M$ denoted by $\mathbf{H}_M$ is thus simply comprised of the corresponding rows and columns in the fully dense version $\mathbf{H}$. Hence, we only have to compute $\mathbf{H}$ (which is actually the same for all rows) once, from which we can then extract the rows and columns corresponding to $M$ as needed.

Critically, this trick is *not* applicable to the inverse, as $(\mathbf{H}_M)^{-1} \neq (\mathbf{H}^{-1})_M$. However, using the fact that the removal of one parameter $p$ simply drops the corresponding row and column from $\mathbf{H}$, we can actually update the inverse to remove parameter $p$ directly using a single step of Gaussian elimination, with cost $\Theta(d_{col}^2)$. The following result, whose proof is in the Appendix, formalizes this.

**Lemma 1** (Row & Column Removal). *Given an invertible $d_{col} \times d_{col}$ matrix $\mathbf{H}$ and its inverse $\mathbf{H}^{-1}$, we want to efficiently compute the inverse of $\mathbf{H}$ with row and column $p$ removed, which we denote by $\mathbf{H}_{-p}$. This can be accomplished through the following formula:*

$$\mathbf{H}_{-p}^{-1} = \left(\mathbf{H}^{-1} - \frac{1}{[\mathbf{H}^{-1}]_{pp}}\mathbf{H}_{:,p}^{-1}\mathbf{H}_{p,:}^{-1}\right)_{-p},\tag{4}$$

*which corresponds to performing Gaussian elimination of row and column $p$ in $\mathbf{H}^{-1}$ followed by dropping them completely. This has $\Theta(d_{col}^2)$ time complexity.*

The resulting pseudocode is shown in Algorithm 1, where we avoid constantly resizing $\mathbf{H}^{-1}$ (and correspondingly changing indices) by utilizing the fact that row and column $p$ have no effect on any future calculations after they have been eliminated by Lemma 1 as they are 0 (and the non-zero diagonal element is never accessed again). One can check that this algorithm applies OBS to a single row of $\mathbf{W}$ with a per-step cost of $\Theta(d_{\text{col}}^2)$, and thus $\Theta(k \cdot d_{\text{col}}^2)$ overall time for pruning $k$ weights.

---

**Algorithm 1** Prune $k \leq d_{\text{col}}$ weights from row $\mathbf{w}$ with inverse Hessian $\mathbf{H}^{-1} = (2\mathbf{XX}^\top)^{-1}$ according to OBS in $O(k \cdot d_{\text{col}}^2)$ time.

---

$M = \{1, \ldots, d_{\text{col}}\}$
**for** $i = 1, \ldots, k$ **do**
$\quad p \leftarrow \text{argmin}_{p \in M} \frac{1}{[\mathbf{H}^{-1}]_{pp}} \cdot w_p^2$
$\quad \mathbf{w} \leftarrow \mathbf{w} - \mathbf{H}_{:,p}^{-1} \frac{1}{[\mathbf{H}^{-1}]_{pp}} \cdot w_p$
$\quad \mathbf{H}^{-1} \leftarrow \mathbf{H}^{-1} - \frac{1}{[\mathbf{H}^{-1}]_{pp}}\mathbf{H}_{:,p}^{-1}\mathbf{H}_{p,:}^{-1}$
$\quad M \leftarrow M - \{p\}$
**end for**

---

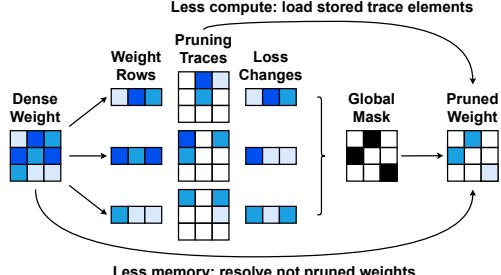

Figure 1: Efficient global OBS using the row-wise results.

**Step 2: Jointly Considering All Rows.** Applying the OBS framework to the full weight matrix $\mathbf{W}$ rather than just to each row independently requires fast access to all $d_{\text{row}}$ row-wise inverse Hessians, in order to select the weight with the smallest overall pruning score in each step. However, storing $d_{\text{row}}$ matrices of size $d_{\text{col}} \times d_{\text{col}}$ each in GPU memory can be too expensive; while it would be possible to offload some Hessians to main memory, this could result in a large number of expensive memory transfers. However, since there is no Hessian interaction between rows, the final compressed weights of each row only depend on the total number of parameters that were pruned in it. Similarly, the change in loss incurred by pruning some weight only depends on the previously pruned weights in the same row, which also means that the order in which weights are pruned in each row is fixed.

The consequence of these insights is that we can process each row independently, pruning all weights in order while always recording the corresponding change in loss $\delta\mathcal{L}_p = w_p^2/[\mathbf{H}^{-1}]_{pp}$. At the end, we know $\delta\mathcal{L}_p$ for all $d$ weights and can then simply determine the global mask that would be chosen by OBS on the full matrix by selecting the weights with the lowest values in order, requiring only $\Theta(d)$ extra memory. We note that once the per-row masks $M_i$ are known, we can directly solve for the optimal update of the remaining weights via the corresponding group OBS formula [21] $\boldsymbol{\delta}_{M_i} = \mathbf{H}_{:,M_i}^{-1}((\mathbf{H}^{-1})_{M_i})^{-1}\mathbf{w}_{M_i}$. This will be considerably faster in practice than simply rerunning the iterative pruning process in Algorithm 1. Alternatively, if enough CPU memory is available, one can keep the full *pruning trace* of each row, that is, the full weight vector after every individual pruning step, in CPU memory and ultimately simply reload the entries corresponding to the global mask. This requires $O(d_{\text{row}} \cdot d_{\text{col}}^2)$ extra CPU memory but avoids a second computation pass to reconstruct the not pruned weights and will therefore be faster. Figure 1 visualizes both options just discussed.

**Implementation Details.** In practice, the matrix $\mathbf{H}$ might not always be invertible for reasons such as using too few data samples or dead / linearly dependent inputs. The former can usually be addressed by extending the calibration dataset with augmentations (additional augmented samples only need to be accumulated into the Hessian once and are thus very cheap to include) and the latter can be prevented by adding a small diagonal dampening term to the Hessian before inverting it. Second, a direct GPU implementation of Algorithm 1 will perform a large number of small CUDA calls, which can be expensive. This overhead can be removed by using batch operations to process multiple matrix rows simultaneously—for more details please see our sample implementation. Finally, when applied to an already sparse weight matrix, the complexity of our algorithm can scale cubically with the row-density by working with a dense version of the weights / Hessians consisting only of the non-zero elements and mapping the pruning result back at the end.

**N:M Sparsity.** Our method can be easily extended to various forms of *semi-structured* sparsity. This includes, for example, the N:M sparsity pattern [45], which enforces exactly $N$ non-zero values in each block of $M$ consecutive weights, and is becoming popular due to support on newer NVIDIA hardware [30]. Adapting our algorithm to this pattern requires only one simple change: instead of selecting the weight with the smallest change in loss, we select the weight with the smallest change in loss that is in a block with $< N$ pruned weights. We note that all rows have exactly the same sparsity $1 - N/M$ in the N:M pattern and so we can terminate per-row pruning as soon as this target sparsity value is reached. For the same reason, there is no need for the global mask selection step described earlier. Thus, our method will be even more efficient in this case.

**Block-Sparsity.** Another practically relevant pruning pattern, particularly in the context of CPU acceleration [7, 22], is *block-pruning*, where zeros appear only in consecutive blocks of size $c$, which is typically a small number like 4 or 8. We follow recent work [21] that extends the OBS framework to pruning small groups of connected weights in order to account for the correlation between them, using the following formulas for the target block and weight update, respectively:

$$\mathbf{w}_P = \operatorname{argmin}_{\mathbf{w}_P} \mathbf{w}_P^\top((\mathbf{H}^{-1})_P)^{-1}\mathbf{w}_P, \quad \boldsymbol{\delta}_P = -\mathbf{H}_{:,P}^{-1}((\mathbf{H}^{-1})_P)^{-1}\mathbf{w}_P, \tag{5}$$

where $P$ denotes the set of indices corresponding to one block. Algorithm 1 can easily be adapted to operate on blocks using the above equations and applying the update of $\mathbf{H}^{-1}$ via Lemma 1 successively for all $p \in P$. Although there are now only $d_{col}/c$ steps per row, each update of $\mathbf{H}^{-1}$ also takes $O(c \cdot d_{\text{col}}^2)$ time and so the overall asymptotic runtime stays the same. Additional practical overhead only comes from the extra $O(c^2 \cdot d_{\text{col}}^2)$ terms that are the result of computing and multiplying with the $c \times c$ matrices $((\mathbf{H}^{-1})_P)^{-1}$.

# 5 The Optimal Brain Quantizer (OBQ)

Although the classical OBS framework [23, 13] has inspired a long line of work on pruning methods for DNNs [39, 10, 27], so far it has not been used for quantization. We now show that our results from the previous section can in fact be extended to quantization in an effective and accurate way, via a method which we call the Optimal Brain Quantizer (OBQ), in the spirit of [23, 13].

**The Quantization Order and Update Derivations.** Under the standard assumption that the gradient at the current point $\mathbf{w}$ is negligible, the OBS formulas for the optimal weight to be pruned $w_p$ and the corresponding update $\boldsymbol{\delta_p}$ can be derived by writing the locally quadratic problem under the constraint that element $p$ of $\boldsymbol{\delta_p}$ is equal to $-w_p$, which means that $w_p$ is zero after applying the update to $\mathbf{w}$. This problem has the following Lagrangian:

$$L(\boldsymbol{\delta_p}, \lambda) = \boldsymbol{\delta_\mathbf{p}^\top} \mathbf{H} \boldsymbol{\delta_p} + \lambda(\mathbf{e_p^\top} \boldsymbol{\delta_p} - (-w_p)), \tag{6}$$

where $\mathbf{H}$ denotes the Hessian at $\mathbf{w}$ and $\mathbf{e_p}$ is the $p$th canonical basis vector. The optimal solution is then derived by first finding the optimal solution to $\boldsymbol{\delta_p}$ via setting the derivative $\partial L / \partial \boldsymbol{\delta_p}$ to zero and then substituting this solution back into $L$ and solving for $\lambda$; please see e.g. [13, 39] for examples.

Assume a setting in which we are looking to quantize the weights in a layer on a fixed grid of width $\Delta$ while minimizing the loss. To map OBS to a *quantized* projection, we can set the target of the Lagrangian constraint in (6) to $(\text{quant}(w_p) - w_p)$, where $\text{quant}(w_p)$ is the weight rounding given by quantization; then $w_p = \text{quant}(w_p)$ after the update.

Assuming we wish to quantize weights iteratively, one-at-a-time, we can derive formulas for the "optimal" weight to quantize at a step, in terms of minimizing the loss increase, and for the corresponding optimal update to the unquantized weights, in similar fashion as discussed above:

$$w_p = \text{argmin}_{w_p} \frac{(\text{quant}(w_p) - w_p)^2}{[\mathbf{H}^{-1}]_{pp}}, \quad \boldsymbol{\delta_p} = -\frac{w_p - \text{quant}(w_p)}{[\mathbf{H}^{-1}]_{pp}} \cdot \mathbf{H}_{:,p}^{-1}. \tag{7}$$

In fact, since $-w_p$ is a constant during all derivations, we can just substitute it with $(\text{quant}(w_p) - w_p)$ in the final result. We note that the resulting formulas are a generalization of standard OBS for pruning, if $\text{quant}(\cdot)$ always "quantizes" a weight to 0, then we recover the original form.

**Quantizing Full Layers.** At first glance, OBQ might appear curious since one usually quantizes *all* weights in a layer, leaving no more weights to update. At the same time, the weight selection metric influences only the quantization order, but not the quantization value. However, this view changes when considering OBQ in the context of our efficient one-weight-at-a-time pruning algorithm described in the previous section. Specifically, using OBQ, we can greedily quantize the currently "easiest" weight by the above metric, and then adjust all the remaining unquantized weights to compensate for this loss of precision, thus changing their value. We then choose the next weight to quantize, and so on. This can result in quantization assignments that are different from the ones that would have been chosen by rounding initially, and in better overall quantization results. Concretely, to realize this, we can plug (7) into Algorithm 1 in order to iteratively quantize weights for a given layer, leading to the similar Algorithm in the Appendix, thus essentially unifying pruning and quantization.

**Quantization Outliers.** One practical issue with this greedy scheme can occur especially when applied to quantization grids that permit some outliers in order to achieve a lower error on the majority of weights, which are currently standard [4, 34]. Since these outliers can have high quantization error, they will usually be quantized last, when there are only few other unquantized weights available that may be adjusted to compensate for the large error incurred by quantizing the outliers. This effect can become worse when some weights are pushed even further outside the grid by intermediate updates. We prevent this with a simple but effective heuristic: we quantize outliers, e.g. weights with a quantization error $> \Delta/2$ where $\Delta$ is the distance between quantized values, as soon as they appear (which typically happens only a few times per layer). With this heuristic, OBQ yields a highly effective layer-wise quantization scheme, as our experiments in the next section demonstrate. Finally, we note that the OBQ version of the techniques discussed in Section 4 has all the same runtime and memory characteristics (barring the global step in Figure 1, which is unnecessary for quantization).

# 6 Experiments

**Objectives, Models & Datasets.** To demonstrate the effectiveness and flexibility of our method, we

consider several different standard *post-training compression* scenarios [31, 19, 18]. We begin with settings where only a single type of compression is applied: concretely, we consider unstructured pruning for given FLOP targets, global 2:4 and 4:8 pruning, as well as uniform weight quantization. Additionally, we also study two practical tasks that feature joint pruning and quantization: a GPU scenario where quantization and N:M pruning are combined, as well as a CPU scenario combining quantization and block pruning. We work with variants of the following models and tasks: ResNet [14] for image classification on Imagenet [38], YOLOv5 [20] for object detection on COCO [26] and BERT [5] for question answering on SQuAD [36]. Our smaller BERT models denoted by BERT3 and BERT6 correspond to the smaller 3 and 6 layer variants of BERT-base, respectively, trained by [21]. The Appendix contains additional experiments as well as runtime information of our algorithms.

**Experimental Setup.** All of our calibration datasets consist of 1024 random training samples. For ImageNet, where we use roughly $0.1\%$ of the training data, we additionally apply standard flipping and cropping augmentations to artificially increase the size of this dataset by $10\times$; other tasks do not use any augmentations. While the effect of augmentations is typically minor, they are very cheap to include for our method. For ResNet models, batchnorm statistics are reset using 100 batches of 128 samples from the calibration set with standard augmentations. For other models, we apply mean and variance correction [32, 1] after all normalization layers (so that the correction parameters can be easily merged and incur no extra cost) on a single batch of samples of size 128 (for YOLO) and 512 (for BERT). We found this to be more effective than batchnorm tuning for YOLO, and the BERT models have no batchnorm layers.

When compressing to a given FLOP or timing constraint, we need to solve the problem of identifying per-layer compression targets, which match the constraint, while maximizing accuracy. To identify these non-uniform targets, we follow the approach of [10]: we first collect a "model database" containing for each compression level (e.g. bit-width or sparsity setting) the corresponding (independently) compressed version of each layer. For building a joint sparse and quantized database we simply sparsify layers first and then apply quantization to the remaining weights. Next, similarly to [19], we compute the layer-wise calibration losses (without augmentations) for all compression levels, corresponding to the models with exactly one layer compressed to a certain level. Then, given layer-wise FLOP or timing information, we set up a constrained layer-wise compression problem of the form described in AdaQuant [19] and solve it with the dynamic programming algorithm of SPDY [10]. This returns an optimal per-layer assignment of compression levels, for which we can then easily produce the corresponding model, via a two-step process: we first stitch together layers at the corresponding compression levels from the database, and then perform the discussed statistics correction to recover some extra accuracy [19].

**Unstructured Sparsity.** We begin our experiments with *unstructured* sparsity, comparing against global magnitude pruning (GMP) [46], the approximate layer-wise OBS method L-OBS [6], and the post-training pruning state-of-the-art method AdaPrune [18]. As a sanity check, we examine in Figure 2 whether our method provides better results in terms of layer-wise squared error, pruning the first layer of a ResNet18 (RN18) model to several sparsities. In this metric, ExactOBS performs best by a wide margin ahead of AdaPrune, which significantly outperforms the other two methods.

Next, in Table 1, we turn our attention to the practical problem of pruning various models to achieve a given FLOP reduction of $2\times-4\times$, applying the per-layer target sparsity optimization technique described above. Our ExactOBS generally performs best (except for YOLOv5l $2\times$ where all methods perform similarly in terms of mAP@0.5) and at $4\times$ FLOP reduction even with a $> 1\%$ gap to the next best method. Interestingly, on the hard-to-prune BERT model, ExactOBS appears to be the only method which still produces reasonable results at higher reduction targets. For BERT $3\times$ and $4\times$, where the performance drop of all methods is $> 2\%$, we additionally assess the compatibility of our results with the more powerful (but also more expensive) post processing method *global AdaPrune* [10]. While this global optimization technique is able to recover lost accuracy, the ExactOBS models still maintain a $> 0.5\%$ and $> 2\%$ F1 advantage, respectively (see Table 5).

**N:M Sparsity.** Next, we study the performance of our method for *semi-structured* sparsity via the N:M pattern. Specifically, we compare against the 4:8 results of AdaPrune with batchnorm tuning [18] on ResNet models (see Table 2) in addition to a 2:4 comparison on BERT models (see Table 3). We highlight that ExactOBS matches or even slightly exceeds the 4:8 results of AdaPrune with the considerably more stringent 2:4 pattern, which is already well supported on NVIDIA hardware. Furthermore, in a 2:4 comparison on BERT models, ExactOBS achieves 1–2% higher F1 scores.

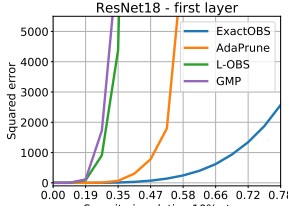

Figure 2: RN18 squared error.

| Method | ResNet50 – 76.13 | | | YOLOv5l – 66.97 | | | BERT – 88.53 | | |
|---|---|---|---|---|---|---|---|---|---|
| | 2× | 3× | 4× | 2× | 3× | 4× | 2× | 3× | 4× |
| GMP | 74.86 | 71.44 | 64.84 | 65.83 | 62.30 | 55.09 | 65.64 | 12.52 | 09.23 |
| L-OBS | 75.48 | 73.73 | 71.24 | **66.21** | 64.47 | 61.15 | 77.67 | 3.62 | 6.63 |
| AdaPrune | 75.53 | 74.47 | 72.39 | 66.00 | 64.88 | 62.71 | 87.12 | 70.32 | 18.75 |
| ExactOBS | **75.64** | **75.01** | **74.05** | 66.14 | **65.35** | **64.05** | **87.81** | **85.87** | **82.10** |

Table 1: Unstructured pruning for different FLOP reduction targets.

| Model | Dense | AdaPrune 4:8 | ExactOBS 2:4 | ExactOBS 4:8 |
|---|---|---|---|---|
| ResNet18 | 69.76 | 68.63 | 68.81 | **69.18** |
| ResNet34 | 73.31 | 72.36 | 72.66 | **72.95** |
| ResNet50 | 76.13 | 74.75 | 74.71 | **75.20** |

Table 2: Semi-structured N:M pruning (+ batchnorm tuning) of all layers except the first and the last.

| Model | Dense | AdaPrune | ExactOBS |
|---|---|---|---|
| BERT3 | 84.66 | 82.75 | **83.54** |
| BERT6 | 88.33 | 85.02 | **86.97** |
| BERT | 88.53 | 85.24 | **86.77** |

Table 3: Semi-structured 2:4 pruning of all layers except the embeddings.

**Quantization.** Additionally, we compare OBQ's *independent* performance (after batchnorm tuning) with the state-of-the-art *sequential* post-training methods AdaQuant [19], AdaRound [31] and BRECQ [24]. We perform standard asymmetric per-channel quantization of all weights, using the authors' implementations. We rerun all methods on Torchvision [28] ResNets to ensure a uniform baseline. The quantization grids for OBQ as well as AdaRound are determined with the same LAPQ [34] procedure that is used by BRECQ. Surprisingly, we find that, despite optimizing layers independently, OBQ achieves very similar (sometimes even slightly better) accuracies as existing non-independent methods for 4 and 3 bits. This suggests that it should be well-suited for mixed precision applications where one needs to quickly generate many non-uniform models optimized for different constraints. (However, we note that ExactOBS can also be applied sequentially; see Appendix.)

| Method | Lw. | Ind. | ResNet18 – 69.76 | | | ResNet50 – 76.13 | | |
|---|---|---|---|---|---|---|---|---|
| | | | 4bit | 3bit | 2bit | 4bit | 3bit | 2bit |
| AdaRound | yes | no | 69.34 | 68.37 | 63.37 | 75.84 | 75.14 | 71.58 |
| AdaQuant | yes | no | 68.12 | 59.21 | 00.10 | 74.68 | 64.98 | 00.10 |
| BRECQ | no | no | 69.37 | 68.47 | 64.70 | 75.88 | 75.32 | 72.41 |
| OBQ (ours) | yes | yes | 69.56 | 68.69 | 64.04 | 75.72 | 75.24 | 70.71 |

Table 4: Comparison with state-of-the-art post-training methods for asymmetric per-channel weight quantization of all layers. We mark whether methods are Layer-wise (Lw.) or Independent (Ind.).

| Methods | BERT | |
|---|---|---|
| | 3× | 4× |
| gAP + AdaPrune | 86.99 | 84.10 |
| gAP + ExactOBS | **87.57** | **86.42** |

Table 5: Further improving results in Table 1 with > 3% performance drops through more expensive post-processing via global AdaPrune (gAP).

**BOP-Constrained Mixed GPU Compression.** We now consider a practical setting where we are given a trained model together with some calibration data and want to compress this model for efficient inference on an NVIDIA GPU which supports 8-bit and 4-bit arithmetic, also in combination with 2:4 sparsity. Thus, there are 4 possible compression choices per layer: 8bit weights + 8bit activations (8w8a), 4w4a, 8w8a + 2:4 and 4w4a + 2:4. Unlike in the previous section, we do *symmetric* per-channel quantization of the weights as it has better hardware support; activations are quantized asymmetrically per-tensor. We then generate mixed precision configurations for various BOP (number of bits times FLOPs) reduction targets and visualize the resulting compression-accuracy trade-off curves in Figure 3. In summary, at the cost of a ≈ 2.5% relative performance drop, we can achieve a 12 − 14× BOP reduction for ResNets and a 7 − 8× reduction for the more challenging YOLO and BERT models (relative to the compute in compressible layers). To the best of our knowledge, we are the first to consider joint N:M pruning and quantization in a post-training setting. Recent work [3] also studies joint 4w4a + 2:4 compression for ResNet18 but with 90 epochs of (sparse) Quantization-Aware Training (QAT) on the full dataset and report 67.33% accuracy. Although not perfectly comparable (we keep the first layer dense and their dense baseline has 0.94% higher accuracy and uses 4:8 sparse activations), we achieve similar 67.20% accuracy for 4w4a + 2:4 *post training*, which emphasizes the effectiveness of our methods for joint sparsification and quantization.

**Time-Constrained CPU Compression.** Lastly, we explore a similar scenario, but targeting actual CPU inference speedup on a 12-core Intel Xeon Silver 4214 CPU using the DeepSparse inference

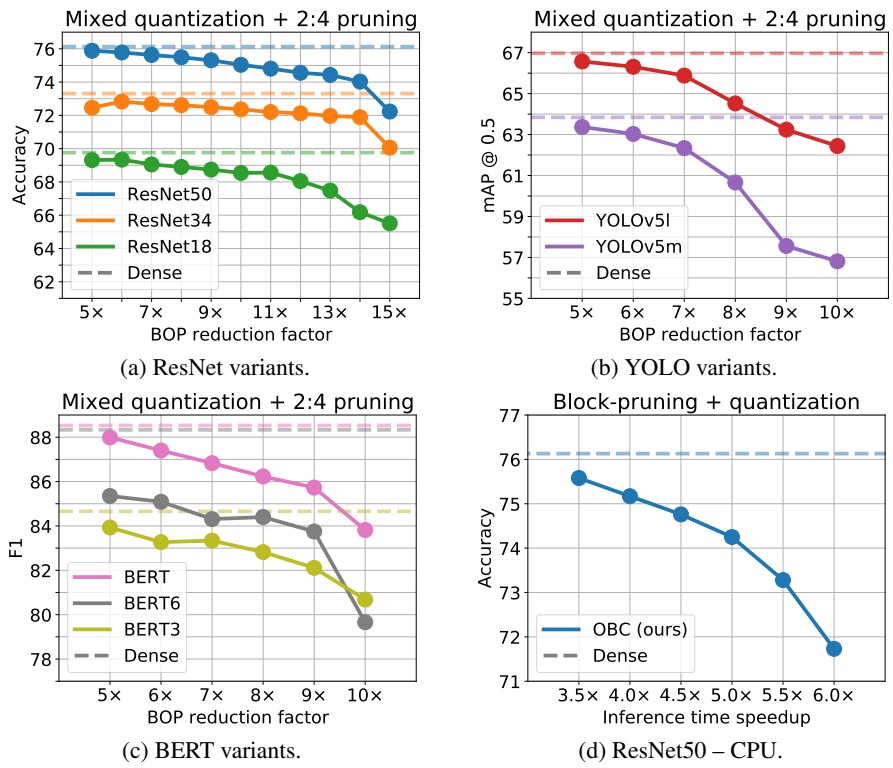

(a) ResNet variants.

(b) YOLO variants.

(c) BERT variants.

(d) ResNet50 – CPU.

Figure 3: (a) to (c) Mixed quantization and 2:4 pruning for various BOP reduction targets. (d) Joint block-pruning and quantization for CPU inference time speedups.

engine [35, 22], which provides acceleration for joint 8-bit quantization and block-sparsity with blocksize 4. In this case, we work with real layer-wise timing data (for batchsize 64), as in [9]. There are 30 available block-sparsity targets per-layer, in steps of pruning 10% of the remaining weights, all of which are further quantized to 8 bits. The base acceleration of the dense 8 bit model is $\approx 2.7\times$ on top of which sparsity speedup acts roughly multiplicatively. Figure 2d shows results for ResNet50 and several (real-time) speedup targets—we achieve $4\times$ and $5\times$ (actual) speedup with $1\%$ and $2\%$ accuracy loss, respectively. These are the first full post-training results in this setting (the authors of [10] only performed 4-block pruning post-training, followed by 5 epochs of QAT on the entire ImageNet dataset), and they show very encouraging accuracy-speedup trade-offs.

## 7   Conclusions & Future Work

We have presented a new efficient and accurate approach for solving the layer-wise compression problem, and built on it to obtain state-of-the-art post-training compression solutions for both pruning and quantization. Our framework should be naturally extensible to *structured* pruning, which in fact should allow for further optimizations, and should also be compatible with further compression via unstructured pruning and quantization. Our results suggest that post-training compression may be able to reach comparable accuracies to much more expensive retraining methods. We plan to investigate this in future work, in particular in the context of more resource-intensive models, such as very large-scale language models.

## 8   Acknowledgements

We gratefully acknowledge funding from the European Research Council (ERC) under the European Union's Horizon 2020 programme (grant agreement No 805223 ScaleML), as well as computational support from AWS EC2. We thank Eldar Kurtic for providing us BERT code and pretrained models, and the Neural Magic Team, notably Michael Goin and Mark Kurtz, for support with their software.

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
