# Contents

## A Appendix

### A.1 Proof of Lemma 1 (Row & Column Removal)

*Proof.* First, we observe that element $j$ in row $i$, i.e. $[\mathbf{A}]_{ij}$, is set to 0 by the equivalent matrix transformation of subtracting $[\mathbf{A}]_{ij}$ times column $i$ denoted by $\mathbf{A}_{:,i}$ divided by the corresponding diagonal element $[\mathbf{A}]_{ii}$ (similarly, elements in column $i$ can be set to 0 by subtracting row $i$). Thus, Lemma 1 corresponds to zeroing $\mathbf{H}_{pi}^{-1}$ and $\mathbf{H}_{ip}^{-1}$ for $i \neq p$ via equivalent matrix transformations, or in other words, Gaussian elimination of one row and column.

Next, we apply these equivalent matrix transformations to both sides of the obvious equality $\mathbf{H}^{-1}\mathbf{H} = \mathbf{I}$, which ultimately gives an equation of the following $\mathbf{AB} = \mathbf{C}$ form:

$$\begin{bmatrix} \mathbf{A_1} & \mathbf{0} & \mathbf{A_2} \\ \mathbf{0}^\top & a & \mathbf{0}^\top \\ \mathbf{A_4} & \mathbf{0} & \mathbf{A_3} \end{bmatrix} \cdot \begin{bmatrix} \mathbf{B_1} & \mathbf{b_1} & \mathbf{B_2} \\ \mathbf{b_4}^\top & b & \mathbf{b_2}^\top \\ \mathbf{B_4} & \mathbf{b_3} & \mathbf{B_3} \end{bmatrix} = \begin{bmatrix} \mathbf{I} & \mathbf{c_1} & \mathbf{0} \\ \mathbf{c_4}^\top & c & \mathbf{c_2}^\top \\ \mathbf{0} & \mathbf{c_3} & \mathbf{I} \end{bmatrix}. \tag{8}$$

Notice now that the entries of $\mathbf{B}$ corresponding to the eliminated row and column in $\mathbf{A}$ do not affect the $\mathbf{I}$ and $\mathbf{0}$ blocks in $\mathbf{C}$ since they are always multipled by 0. Thus, the matrix of the $\mathbf{A_i}$ blocks must be the inverse of the $\mathbf{B_i}$ block matrix, which is exactly what we wanted to calculate. $\qquad\square$

## A.2 ExactOBS Global Step Pseudocode

This section provides more details about the global step of the ExactOBS algorithm described in Section 4 in the form of pseudocode.

---

**Algorithm 2** Let $\mathbf{P}$ be a $d_{\text{row}} \times d_{\text{col}}$ matrix storing the order in which weights are pruned by ExactOBS in each row and let $\mathbf{L}$ be the matrix of the corresponding loss-changes $\delta\mathcal{L}$. Then the following procedure determines the global OBS mask with $k$ pruned weights.

---

$Q = \{(i, 0) \mid 1 \leq i \leq d_{\text{row}}\}$
**for** $k$ times **do**
    $i, j \leftarrow \text{argmin}_{(i,j) \in Q} \quad [\mathbf{L}]_{i(j+1)}$ if $j < d_{\text{col}}$ else $\infty$
    $Q \leftarrow Q - \{(i, j)\}$
    $Q \leftarrow Q \cup \{(i, j + 1)\}$
**end for**
$Q$ contains the number of pruned elements $j$ per row $i$, which together with $\mathbf{P}$ yields the mask.

---

For increased efficiency, the set $Q$ can be implemented, for example, as a min-heap. Finally, we note that the slightly simpler method of picking the $k$ smallest elements in $\mathbf{L}$ and then counting how many were picked in each row typically produces essentially the same results as Algorithm 2 in practice since the loss changes generally increase monotonically as more weights are pruned.

## A.3 OBQ-ExactOBS Algorithm Pseudocode

The OBQ version of the ExactOBS algorithm is given below; we emphasize the similarity to the pruning variant of ExactOBS shown in Algorithm 1.

---

**Algorithm 3** Quantize $k \leq d_{\text{col}}$ weights from row $\mathbf{w}$ with inverse Hessian $\mathbf{H}^{-1} = (2\mathbf{X}\mathbf{X}^{\top})^{-1}$ according to OBS in $O(k \cdot d_{\text{col}}^2)$ time.

---

$M = \{1, \ldots, d_{\text{col}}\}$
**for** $i = 1, \ldots, k$ **do**
    $p \leftarrow \text{argmin}_{p \in M} \frac{1}{[\mathbf{H}^{-1}]_{pp}} \cdot (q(w_p) - w_p)^2$
    $\mathbf{w} \leftarrow \mathbf{w} - \mathbf{H}_{:,p}^{-1} \frac{1}{[\mathbf{H}^{-1}]_{pp}} \cdot (w_p - q(w_p))$
    $\mathbf{H}^{-1} \leftarrow \mathbf{H}^{-1} - \frac{1}{[\mathbf{H}^{-1}]_{pp}} \mathbf{H}_{:,p}^{-1} \mathbf{H}_{p,:}^{-1}$
    $M \leftarrow M - \{p\}$
**end for**

---

## A.4 Further Experiment Details

We now provide some additional details about our experiments in Section 6.

**Bias and Variance Correction.** Although our bias and variance correction step applied to YOLO and BERT models is similar to the schemes described in [32] and [1], we now describe our exact procedure for additional clarity:

1. Sample one batch from the calibration dataset.

2. Perform inference on this batch with the *dense* model and record after each normalization layer the mean $\mu_{\text{dense}}^{\ell}$ and standard deviation $\sigma_{\text{dense}}^{\ell}$ for each channel (for CNNs) / feature (for Transformers) over this batch.

3. Perform inference on this batch with the *compressed* model and record the means $\mu_{\text{comp}}^{\ell}$ and standard deviations $\sigma_{\text{comp}}^{\ell}$ as in step 2, while already applying mean and variance correction to the layer outputs $X^{\ell}$ via:

$$Y^{\ell} = \frac{\sigma_{\text{dense}}^{\ell}}{\sigma_{\text{comp}}^{\ell}} \cdot (X^{\ell} - \mu_{\text{comp}}^{\ell} + \mu_{\text{dense}}^{\ell}) \tag{9}$$

4. Merge (9) into the affine parameters of the respective normalization layer.

We note that it is critical to apply the statistics correction already while computing the compressed means and variances in step 3 in order to properly account for compounding distribution shifts.

**Non-Uniform Sparsity Choices.** The method we use for determining per-layer (unstructured or blocked) sparsity values to reach a certain overall budget with minimal accuracy loss requires a discrete set of sparsity choices per layer. For both unstructured and blocked sparsity, we follow [10] and choose a grid where each point prunes the same fraction of remaining weights $\delta$. Hence, sparsity choice $s_i$ is given by:

$$s_i = 1 - (1 - \delta)^i. \tag{10}$$

In both cases we choose $\delta = 0.9$, which corresponds to pruning 10% of the remaining weights. For unstructured sparsity, we generate choices until $s_i > 0.99$ and for blocked sparsity until $s_i > 0.95$. We note that these sets of sparsity options are chosen to allow for maximum flexibility. However, in many cases, similar results can likely be achieved with significantly fewer, but more carefully selected (e.g. using the fact that very high sparsities will typically never be chosen for lower FLOP reduction targets), options and thus less required database storage.

**Activation Quantization.** In our GPU-focussed quantization + 2:4 pruning experiments we also quantize all activations. This is done by simply optimizing the zero point and quantization scale for one input batch of each layer using exactly the same procedure as for the weights, just on tensor-instead of channel-level (which is the same LAPQ [34] procedure also used by BRECQ [24]). This is again done independently for each layer and the corresponding quantization information is stored in the model database to allow for quick stitching. More advanced schemes such as reoptimizing the weights to better match the quantized inputs (see Appendix A.8) may be possible, but we found the simple procedure just described to already work quite well.

## A.5 Timing Information

In this section, we provide detailed information about the runtime of our method. All numbers reported here are for the execution on a single NVIDIA RTX 3090 GPU using our PyTorch implementations. Pruning runs with a global step are performed with the "less compute" variant described in Figure 1. Hence an entire database of *many* pruning levels can be generated in approximately the time shown for unstructured and block pruning runs here.

**PTQ Runtime Comparison.** We begin with a runtime comparison of existing state-of-the-art post-training methods at the task of quantizating the weights of all layers of a ResNet50 to 4 bits. All timings were collected by executing the authors' open-source implementations on the same hardware, the results are shown in Table 6.

| Model | BitSplit | AdaRound | AdaQuant | BRECQ | OBQ |
|-------|----------|----------|----------|-------|-----|
| ResNet50 | 124m | 55m | 17m | 53m | 65m |

Table 6: Runtimes of post-training quantization methods in minutes (m).

BRECQ, AdaRound and our method OBQ all take around one hour to fully quantize ResNet50, the former two slightly less and the latter slightly more. Meanwhile, BitSplit takes about twice as long, whereas AdaQuant is $3\times$ faster. However, as shown in Table 4 in the main text (as well as in Table 9), AdaQuant is also considerably less accurate than the other methods. In summary, the runtime of ExactOBS is in line with existing post-training methods. Additional optimizations, like periodically shrinking the Hessian by omitting rows/columns of pruned/quantized weights, can likely improve the practical speed further.

**Different Compression Types.** Next, we study the runtime of ExactOBS applied to different types of compression problems. We consider a smaller model (YOLOv5s), a medium model (ResNet50) and a larger one (BERT). The corresponding runtimes for all compression types featured in this work are listed in Table 7.

In general, we can see that quantization and unstructured pruning take about the same time, which matches with the fact that the corresponding algorithms are very similar. Correspondingly, 2:4

| Model | Quant | Unstr | 4-block | 2:4 | Quant 2:4 |
|---|---|---|---|---|---|
| ResNet50 | 65m | 64m | 61m | 31m | 35m |
| YOLOv5s | 7m | 6m | 10m | 3m | 4m |
| BERT | 111m | 103m | 142m | 51m | 56m |

Table 7: Runtimes of ExactOBS for different models and compression types in minutes (m).

pruning and quantizing a 2:4 pruned model are only approximately half as expensive, which is again expected as they perform half the work. For YOLO and BERT, blocked pruning is the most expensive compression type due to the overheads incurred by handling the additional $c \times c$ block matrices (see Section 4). Interestingly, for ResNet50, this is not the case, which is probably related to the highly non-uniform compute distribution that is discussed in more detail in the next paragraph. Overall, these results show that our techniques are quick for small models and still reasonably efficient even for bigger models like BERT, taking less than 2 hours on a single GPU. Finally, we note that ExactOBS is essentially perfectly parallelizable and its runtime can thus scale linearly with the number of available GPUs.

**Per-Layer Runtimes.** Finally, we note that as the time complexity of OBQ implemented via ExactOBS is $O(d_{\text{row}} \cdot d_{\text{col}}^3)$, i.e. cubic in the column dimension, the overall runtime can often be dominated by a few particularly large layers. This is illustrated e.g. by ResNet50 where, as shown in Figure 4, about $75\%$ of the overall runtime is spent in the $3 \times 3$ convolutions of the last block (which have $d_{\text{col}} \approx 4500$ when unfolded), of which there are just 3 in total. Meanwhile, most of the earlier layers are quantized within seconds. This means that one could, in many cases, reduce the overall compression runtime significantly by applying a faster but less accurate method to just those few bottleneck layers while still achieving more accurate compression on all the others through our techniques.

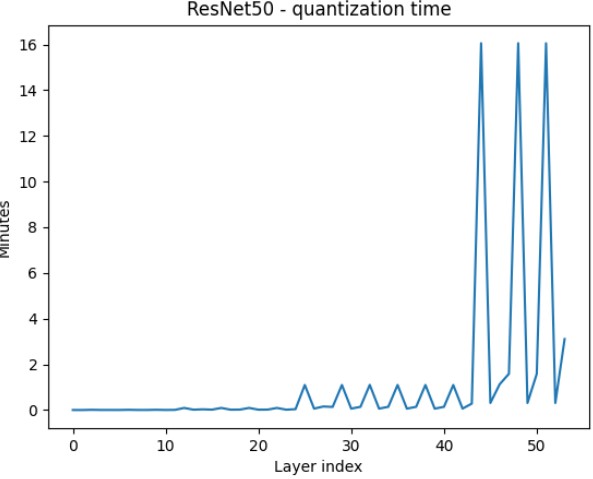

Figure 4: Runtime of OBQ for each layer of ResNet50.

## A.6 Multiple AdaPrune Iterations

While AdaPrune [18] determined all weights to prune in a single step, the authors of [10] found that iterating this process in smaller steps can often improve performance significantly, at quickly increasing computational costs. Our method realizes the very limit of this scheme with one step for each weight. In this section, we study how OBQ comares against AdaPrune with a varying number of pruning and full reoptimization steps. For that purpose, we prune BERT to uniform $75\%$ sparsity by applying AdaPrune in $k = 2^i$ steps that, as suggested by [10], all prune the same fraction of remaining weights.

| Model | Sparse | ExactOBS | AP $1\times$ | AP $2\times$ | AP $4\times$ | AP $8\times$ | AP $16\times$ |
|-------|--------|----------|-----|-----|-----|-----|------|
| BERT | 75% | **-7.69** | -61.54 | -31.67 | -19.73 | -18.16 | -14.89 |

Table 8: Comparing F1 drops against AdaPrune (AP) with a varying number of pruning/reoptimization steps.

Our results confirm the finding of [10] that iterating AdaPrune multiple times can significantly improve results, as we see the F1 drop decreasing quickly with just a few such "recomputations". Nevertheless, even after 16 full iterations, which have an overall runtime comparable to ExactOBS, the accuracy drop for the (iterative) AdaPrune model is still almost $2\times$ larger than the one of ExactOBS, clearly demonstrating the benefit of our method.

## A.7 Independent Quantization Comparison

In our uniform quantization experiments in the main paper (see Table 4), we only compared OBQ with state-of-the-art *sequential* methods as those are generally significantly more accurate than *independent* ones. However, for completeness, we now additionally compare OBQ with two other methods that have also been used for *independent* layer-wise quantization: BitSplit [6] and AdaQuant [19]. Here we consider symmetric per-channel quantization as this is the quantization mode BitSplit was designed for. Additionally, we compare "raw" quantization performance, that is directly after independent compression without any additional statistics corrections. The results of the comparison are summarized in Table 9.

| Method | ResNet18 – 69.76 | | | ResNet34 – 73.31 | | | ResNet50 – 76.13 | | |
|--------|------|------|------|------|------|------|------|------|------|
| | 4bit | 3bit | 2bit | 4bit | 3bit | 2bit | 4bit | 3bit | 2bit |
| BitSplit | 67.58 | 59.25 | 07.36 | 71.63 | 64.91 | 26.62 | 74.94 | 71.76 | 07.31 |
| AdaQuant | 65.45 | 49.29 | 00.87 | 69.49 | 56.10 | 00.84 | 72.79 | 53.06 | 00.13 |
| OBQ (ours) | **69.18** | **67.14** | **48.34** | **72.85** | **71.01** | **51.62** | **75.50** | **73.61** | **46.33** |

Table 9: Uniform symmetric per-channel weight quantization.

As expected, OBQ clearly outperforms the other two independent methods on all considered models and bitwidths; at 3 bits by several percent in accuracy and at 2 bits it is the only method that does not break down completely without any statistics correction.

## A.8 Sequential Quantization with OBQ

While we primarily focus on the *independent* application of OBC which enables quick stitching of various mixed-compression models, it is also possible to apply OBC sequentially, in similar fashion to state-of-the-art post-training quantization works [31, 19, 24]. While other methods simply perform the per-layer optimization by swapping out the dense model inputs $X_{\text{dense}}$ for the corresponding inputs in the compressed model $X_{\text{comp}}$, this does not suffice for OBQ. If the Hessian is computed on $X_{\text{comp}}$, then the initial dense weights are not a local minimum (with 0 gradient) anymore, hence violating a key assumption of OBQ. Fortunately, this problem can be easily resolved by reoptimizing the dense weights for the new inputs via the closed form solution of linear regression $\mathbf{W}^\top = (\mathbf{X}\mathbf{X}^\top)^{-1}\mathbf{X}\mathbf{Y}^\top$, after which the gradient is 0 again, and OBQ can be applied correctly. We note that $\mathbf{X}\mathbf{Y}^\top$ is a $d_{\text{col}} \times d_{\text{row}}$ matrix which can be easily accumulated over multiple batches similar to the OBQ Hessian $2\mathbf{X}\mathbf{X}^\top$, without any major increase in memory consumption.

As a demonstration, we apply sequential OBQ to the task of quantizating ResNet18 to various bitwidths (in the same setup as in Table 4 in the main paper) and report the results in Table 10. Interestingly, for 4 and 3 bits, the results are essentially the same as for the independent version (with batchnorm statistics correction); only for the 2 bits setting there seems to be a noticeable benefit, catching up with the corresponding BRECQ result. A more detailed investigation of this phenomenon could be a interesting direction for future work.

| Method | ResNet18 – 69.76 | | |
| --- | --- | --- | --- |
| | 4bit | 3bit | 2bit |
| AdaRound | 69.34 | 68.37 | 63.37 |
| AdaQuant | 68.12 | 59.21 | 00.10 |
| BRECQ | 69.37 | 68.47 | 64.70 |
| OBQ + BNT | 69.56 | 68.69 | 64.04 |
| OBQ – sequential | 69.56 | 68.68 | 64.93 |

Table 10: Comparison with sequential OBQ.

## A.9 Impact of ImageNet Data Augmentations

As described in the main submission text, for ImageNet experiments, we expand our calibration set with standard data augmentations by a factor of 10. The is mainly done to ensure that the $2048 \times 2048$ Hessian corresponding to the fully-connected layer of ResNet50 is full rank (which is not the case for just 1024 images) and thus avoid any hyper-parameter tuning of a dampening constant. Additionally, it should serve as a demonstration that augmentations are cheap to use in conjunction with our method, which is not the case for other post-training methods that would require either considerably increased memory (storing many more activations) or runtime (performing full inference on the entire model for each batch in the per-layer optimization).

We now study the impact of these augmentations on our results, for which rerun OBQ (in the setup of Table 4 without them, but using dampening $\lambda = 1$ (relative to the values in the Hessian this is actually a rather small constant) for the last layer of ResNet50. A comparison with the original results is shown in Table 11.

| Method | ResNet18 – 69.76 | | | ResNet50 – 76.13 | | |
| --- | --- | --- | --- | --- | --- | --- |
| | 4bit | 3bit | 2bit | 4bit | 3bit | 2bit |
| OBQ | 69.56 | 68.69 | 64.04 | 75.72 | 75.24 | 70.71 |
| OBQ – no aug | 69.59 | 68.51 | 63.87 | 75.87 | 75.06 | 70.51 |

Table 11: The impact of data augmentations.

As can be seen, the difference between using and not using data augmentations is generally only rather minor at $\approx 0.1 - 0.2\%$. Nevertheless, augmentations are very cheap to use in conjunction with our methods (they only need to be accumulated into the initial per-layer Hessians once) and at the same time avoid a dampening hyper-parameter in several cases; therefore we use them in our ImageNet experiments.

## A.10 Sensitivity to Random Seeds

For a fixed calibration dataset, the ExactOBS algorithm is deterministic. For ResNet models, small amounts of additional randomness are added by the data augmentations that are applied to the calibration dataset as well as by batchnorm tuning which happens with randomly sampled batches; for the other models we consider, there is no extra randomness beyond the initial sampling of the calibration dataset. To assess how much the results of our methods are affected by these random factors, we quantize ResNet18 to 4bit (symmetric per-channel) and prune ResNet50 to the 2:4 pattern, for 5 different seeds each, and report mean standard deviation in Table 12.

| ResNet18 – 4bit | ResNet50 – 2:4 |
| --- | --- |
| $69.28 \pm 0.07$ | $74.74 \pm 0.05$ |

Table 12: Typical random variation of OBC results.

In conclusion, the variation of results with respect to random seeds is generally very low, here $< 0.1\%$, which is in line with other post training methods [31, 24].

## A.11 Compound Compression Comparisons

In the main paper, we focused on independent comparisons for quantization and pruning since existing methods are generally only designed for a single compression approach. In this section, we additionally provide compound comparisons for our GPU and CPU scenarios which combine sparsity and quantization. In particular, we construct a strong baseline by substituting OBC in our mixed setup with the best independent layer-wise pruning and quantization methods, AdaPrune and AdaQuant, respectively. We now provide detailed comparisons for all experiments of Figure 3 from the main text, in Figures 5, 6 and 7.

In summary, it appears that, as expected, the accuracy improvements for the individual compression types shown by the experiments in Section 6 also transfer to the combined setting. More concretely, for the reduction target ranges highlighted in the main paper, that is $12 - 14\times$ for ResNet models and $7 - 8\times$ for others, there is a consistent $0.5 - 1.5$ point gap between OBC and the AdaPruneQuant baseline. For lower BOP reduction / inference time speedup targets, the gap is typically smaller, which is expected as only the less sensitive layers have to compressed more than to the generally very easy 8-bit level. In contrast, the gaps are largest for the highest targets that also require high compression of sensitive layers as this is where the effects of OBC's more accurate layer-wise compression become particularly noticeable.

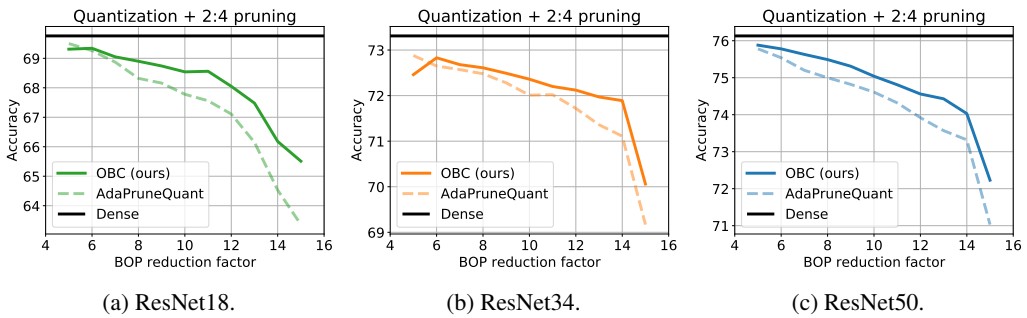

(a) ResNet18.        (b) ResNet34.        (c) ResNet50.

Figure 5: Mixed quantization and 2:4 pruning for various BOP reduction targets on ResNet models.

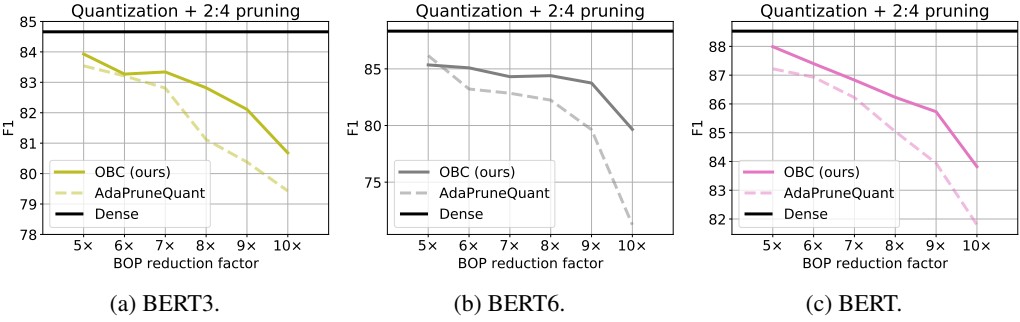

(a) BERT3.        (b) BERT6.        (c) BERT.

Figure 6: Mixed quantization and 2:4 pruning for various BOP reduction targets on BERT models.

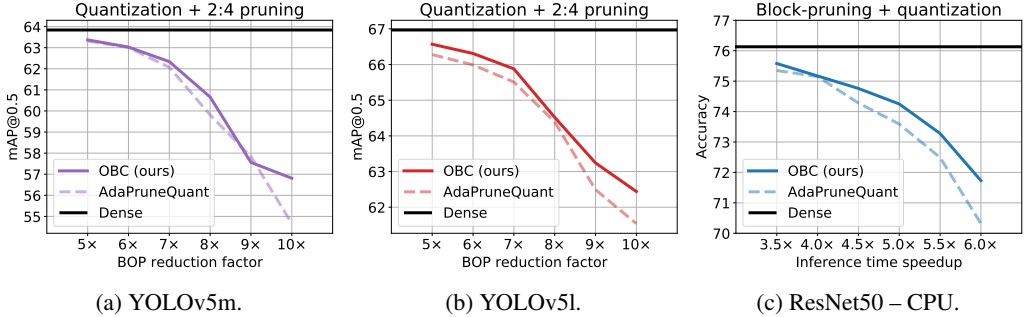

(a) YOLOv5m.

(b) YOLOv5l.

(c) ResNet50 – CPU.

Figure 7: (a) & (b): Mixed quantization and 2:4 pruning for various BOP reduction targets on YOLO models. (c) Block sparsity & quantization for real-time CPU inference speedup targets on ResNet50.