# OpenReview forum: "Optimal Brain Compression: A Framework for Accurate Post-Training Quantization and Pruning"
_NeurIPS.cc/2022/Conference — NeurIPS 2022 Accept_

### Official Review · Reviewer_CZ4H · 2022-07-01

**Rating:** 6
**Confidence:** 4
**Soundness:** 3 good
**Presentation:** 2 fair
**Contribution:** 2 fair

**Summary:**

This paper introduces a compression framework for post-training quantization and pruning. The paper takes a layer-by-layer compression method to quantize and sparsify model weights and shows that this can be done under a unified setup without retraining a network from scratch. The evaluation on ResNet and BERT models show that the proposed method lead to 2 points drop in accuracy when composing quantization and pruning while obtaining 12x model size reduction.

**Questions:**

Can you provide a more direct comparison of the proposed unified method with the existing method that also combines sparsification and quantization without layer-by-layer compression? For example, the sparsification can be weight magnitude-based pruning or movement pruning [1], and quantization can be quantize-aware training [2].


[1] Sanh et al. "Movement Pruning: Adaptive Sparsity by Fine-Tuning"
[2] Jacob et al. "Quantization and training of neural networks for efficient integer-arithmetic-only inference."

**Ethics Review Area:**

["I don’t know"]

**Limitations:**

Yes.

**Strengths And Weaknesses:**

Strengths:

- The paper explores the combination of sparsification and quantization from an interesting angle: the optimal brain surgeon framework, and introduces two techniques to induce sparsity and quantization in a layer-by-layer manner.
- Promising results on combining quantization and sparsification using layer-wise compression techniques without extensive retraining.

Weaknesses:

- The problem definition of the work is a bit confusing. Usually, the problem of compression has been defined as an optimization problem where the optimization objective is to reduce some metrics such as model size or latency while satisfying certain accuracy constraints (e.g., little or no accuracy loss). However, the paper defines the problem as a layer-wise compression problem, which raises a few concerns. First, it includes the solution space of compression as part of its problem definition, e.g., saying that the compression has to be layer-wise, but layer-by-layer compression is only one out of many ways to perform compression, and there is no clear advantage on why one has to use layer-by-layer compression. Therefore, including it in the problem definition seems to artificially reduce the solution space (e.g., in the evaluation, the paper does not compare its method with many existing compression methods that also combine sparsification and quantization). Second, the goal of the problem is to find a "compressed version of a layer weight that performs similarly to the original weights", but this appears to be an indirect optimization target where the paper does not show whether it has a direct correspondence to the metrics people care in practice (e.g., model size, latency, compression cost). The paper might need to provide more justification on the rationale behind defining the problem this way.

- The paper introduces the challenge of compression from a rather indirect angle. There have been numerous work on both pruning and quantization in recent years that target compressing a model. However, instead of talking about the challenges of state-of-the-art approaches, the paper starts with the challenges in the OBS (optimal brain surgeon) framework, which was published almost 30 years ago, and its variations, but none of those appear to have been established as the state-of-the-art for model compression. In contrast, there are actually quite a few more recent work that also performs quantization and pruning together in a unified manner, such as https://arxiv.org/abs/2101.09671 and https://arxiv.org/abs/1811.01907. Therefore, it would be better to more directly compare and position itself with respect to those existing works. Btw, the paper does not cite nor compare those unified compression frameworks.

- The difference between the paper's proposed post-training compression and existing compression methods is not very clear. Notably, popular and state-of-the-art compression methods do not require full retraining, such as quantize-aware training.  Instead, only a small portion of the training data (e.g., <10%) is used to calibrate the quantization errors, as in https://arxiv.org/pdf/1702.03044.pdf. Similarly, for pruning, it is rarely the case to train a sparse model from scratch and again only a small number of iterations are performed to calibrate the model to recover the accuracy, as in https://arxiv.org/pdf/1709.05027.pdf. Since the proposed method still need training data (a subset of training samples) and refinement/calibration training, it is unclear how fundamentally this method has an advantage over existing compression techniques.


---------------
Post-rebuttal comments

The authors addressed my concerns. I raised my score from 4 to 6 to reflect my post-rebuttal perspective.

---

> ### Author Response · Authors · 2022-08-02
> **Reply to Reviewer CZ4H -- Part 1**
>
> Thank you very much for the detailed comments!
>
> Since the review merges several questions of interest, we split them into separate topics and address them in turn. Further, due to length, our response is split into two separate comments.
>
> > The problem definition is a bit confusing ...
>
> Our paper is aimed at the post-training compression setting, and follows the same data and time constraints as the significant prior work on post-training compression, e.g. [40, 30, 19, 18, 6]. Specifically, this setting is very restrictive both in terms of data limits (e.g., ~1K samples on ImageNet, < 0.1% of all training data), and computation (e.g., a few hours of computation on a single GPU for the actual compression).
>
> This setting is standard in the post-training compression literature, and is motivated by practical constraints. For instance, the closed division of the MLPerf industry benchmark adopts an identical setup, and the industry-standard TensorRT inference engine performs compression-calibration in the same context. Further, we note that the reference [Liang et al.] which you pointed out discusses this setting as well, under the term “calibration-based post-training.” In particular, in Section 4.3 the authors discuss that many deployment frameworks offer such post-training quantization features, highlighting again its practical importance.
>
> > [The work restricts] the solution space of compression as part of its problem definition, e.g., saying that the compression has to be layer-wise ...
>
> The very restrictive nature of the post-training compression setting leads virtually all state-of-the-art post-training compression methods, e.g. BRECQ, AdaRound, AdaQuant, AdaPrune or BitSplit, to perform compression layer-by-layer. Essentially all of these approaches start from variants of Equation (1) in our paper and compress models either layer-by-layer, or in small blocks of consecutive layers, using increasingly sophisticated algorithms. Thus, the general compression problem is essentially “reduced” to layer-wise compression in the post-training setting. This indirect layer-wise fitting approach is believed to be inherent for post-training methods: finetuning or retraining the full model with just 1K available samples can lead to overfitting, affecting the compressed model’s generalization performance.
>
> In this context, our paper shows that, despite significant prior work on the layer-wise fitting problem, there is still significant room for improvement and that this directly leads to better final model accuracy.
>
> We hope this explains why we focused on this “restricted” version of the compression problem in our text. Nevertheless, we agree that this can be a bit confusing for readers unfamiliar with post-training methods and will make revisions to make it clearer why we particularly consider layer-wise compression.
>
> As for the connection to the OBS framework: our approach to finding better solutions to the layer-wise compression problem is applying an exact version of the OBS framework, hence we discuss all the challenges involved in realizing such a method in detail.

---

> > ### Author Response · Authors · 2022-08-02
> > **Reply to Reviewer CZ4H -- Part 2**
> >
> > > The difference between the paper's proposed post-training compression and existing compression methods is not very clear.
> >
> > Thank you for pointing out this set of interesting references. We will now respond to each one separately, and will add citations where appropriate.
> >
> > **[Zhou et al.]** — To our reading, this paper uses 10% - 30% of the original model’s training iterations but still operates on 100% of the original dataset. This method would therefore fall under fine-tuning methods. By contrast, we use only 0.1% of the training data, which would not allow a fair comparison.
> >
> > Even if we attempt a direct comparison, we note that we achieve 69.56 Top-1 validation accuracy for one-shot 4-bit quantization on ResNet18, and 75.72 Top-1 accuracy on ResNet50. By contrast, this method reports 68.98 Top-1 on ResNet18, and 74.81 Top-1 on ResNet50 for 5-bit quantization. While this comparison is not very precise, as the quantization schemes and the baseline models are not equivalent, we note that we obtain higher accuracy based on the standard Torchvision baseline, without any retraining.
> >
> > Nevertheless, we found this work very interesting, as it is actually complementary to the techniques we propose. One could substitute their simple weight selection and rounding heuristics by our more accurate OBQ method, while retaining their iterative partial quantization and finetuning loop; we will note this in the future work section.
> >
> > **[Ye et al.]** — This paper presents an ADMM-based method (with various additional post-processing heuristics) that performs extensive retraining on the full dataset. Thus, it is not a post-training method. Further, we note that our claim of unification is in terms of using essentially the same algorithmic method both for pruning and quantization, rather than just combining quantization and pruning into a single system.
> >
> > **[Wen et al.]** — This work seems to fully retrain their models with a special group-lasso penalty; only after this process, they can be pruned in one-shot. Again, this does not fit into the post training setting, as essentially the original training loop has to be re-executed with the regularizer.
> >
> > **[Liang et al.]** — While this paper is a detailed survey on both sparsification and quantization methods, it was not clear to us, even after careful reading, where exactly it discusses unified/combined pruning and quantization approaches in particular.
> >
> > > Comparison with retraining methods:
> >
> > Lastly, although this is beyond the scope of the present work, we note that accurate post-training methods can also be interesting when combined with retraining. To illustrate this, we produced 80% and 90% sparse BERT SQuAD models using OBC (in one shot, a gradual application may bring further improvements) and then finetuned them, in the same setting as Movement Pruning [Sanh et al.], with output distillation, for 8 epochs. We achieved 88.10 and 86.55 F1 score, respectively; for comparison, movement pruning reports ~87 and 84.9 F1 for 80 and 90% sparsity, respectively. Thus, we outperform this standard method, even though our pruning is performed initially, in one-shot, and not iteratively. Therefore, exploring this extension to gradual pruning + fine-tuning seems a very promising direction for future work.
> >
> > We hope our response has clarified the general setting of our paper and the advantages of our methods, as well as correspondingly our considerations concerning the experimental comparisons and the writing.
> >
> > If any questions remain, we are happy to engage in further discussion!

---

> > > ### Comment · Reviewer_CZ4H · 2022-08-08
> > > **Post-rebuttal respose**
> > >
> > > After reading the authors' response, my concerns are mostly resolved. The paper can be strengthened by better describing the assumptions for post-training compression because the proposed work is more for a data insufficient scenario, whereas for the majority of the practical use cases, including the model/dataset the paper evaluated, the training data is available during model compression and the compression cost is not a major bottleneck.

---

> > > > ### Author Response · Authors · 2022-08-08
> > > > **Improvement Suggestions & Evaluation**
> > > >
> > > > Thank you for your response, we are glad that we managed to address most of your concerns!
> > > >
> > > > Thank you also for your improvement suggestions, which we would like to briefly address:
> > > >
> > > > > the proposed work is more for a data insufficient scenario, whereas for the majority of the practical use cases, including the model/dataset the paper evaluated, the training data is available during model compression
> > > >
> > > > We would first like to clarify the fact that the evaluation we adopted is completely standard in the area of post-training compression. Specifically:
> > > >
> > > > * Most of the prior work on post-training compression (AdaQuant, BRECQ, AdaRound, BitSplit, AdaPrune etc.) examines performance on the same models & datasets. We adopted those benchmarks for a proper comparison.
> > > >
> > > > * The practical motivation behind this choice is the Closed Division of the MLPerf Industrial Inference Benchmark (see https://mlcommons.org/en/inference-datacenter-20), adopted by major ML companies (NVIDIA, Intel, Microsoft, etc.). This is an industry standard, which has categories for each of the benchmarks we examine (e.g. ResNet50/ImageNet, BERT/SQuAD), in exactly the same post-training setup which we assume here.
> > > >
> > > > In addition to data scarcity, another justification for the post-training setting is practicality, since not all end users may be willing (or able) to retrain their model via gradual pruning or quantization-aware training.
> > > >
> > > > > The paper can be strengthened by better describing the assumptions for post-training compression
> > > >
> > > > To address this remaining suggestion, we have performed a minor revision which further clarifies this upfront.
> > > >
> > > > In our view, this appears to fully address your concerns and improvement suggestions; we would therefore gently ask that you consider increasing your score.

---

### Official Review · Reviewer_7fQD · 2022-07-11

**Rating:** 7
**Confidence:** 3
**Soundness:** 3 good
**Presentation:** 3 good
**Contribution:** 2 fair

**Summary:**

This paper tackles the important problem of post-training compression in deep nets, namely pruning and quantization. The paper adopts the Optimal Brian Surgeon framework for the per-layer squared loss objective, and presents efficient and exact solvers for the greedy and sequential Hessian-based pruning algorithm, allowing it to scale to modern deep nets. The authors also propose a quantization variant, Optimal Brain Quantizer, which leverages the same exact and efficient solver and enables accurate post-training quantization. The authors validate the efficacy of their methods across various tasks, networks, and compression strategies, showcasing real hardware benefits.

**Questions:**

Please see Weaknesses above


**Limitations:**

The authors should mention the limitations of their method, ideally in the last section.

**Strengths And Weaknesses:**

Strengths:
- The paper is well written and very easy to follow. Everything was well organized
- The proposed method is simple and scalable, allowing the authors to implement the exact OBS solver at the scale of modern deep nets
- The idea of extending OBS for quantization seems novel to me, and is very interesting
- The experimental results demonstrate that the proposed method is better/competitive with existing post-training methods

Slight Weaknesses:
- The overall novelty of the method is hard to assess. Please correct me if I'm wrong, but my current understanding is that the main contribution is having an efficient and exact OBS solver, but the implementation itself is built on existing/simple ideas. For instance, the theoretical result (Lemma 1) seems to be a direct application of the Woodbury-Sherman-Morrison formula [1] (if so, it should be cited in the text). I like how the paper is written, but I feel some improvements can be made to better highlight the technical contributions, and distinguish it from existing methods/ideas.

- A comprehensive comparison that includes both the setup cost (time to compress), runtime cost (inference time), and task performance (accuracy) is missing in the main text. Such comparisons are vital to improving the quality of the manuscript, and help the reader better asses the performance of the proposed method compared to prior art.

[1] Woodbury, Max A. Inverting modified matrices. Statistical Research Group, 1950.

---

> ### Author Response · Authors · 2022-08-02
> **Reply to Reviewer 7fQD**
>
> Thank you very much for your useful comments and suggestions, which we will now respond to in detail.
>
> > The overall novelty of the method is hard to assess ...
>
> Thank you for the opportunity to clarify this point. We believe the following algorithmic elements are new:
>
> * The general approach, i.e. the **exact** Optimal Brain Surgeon applied to layer-wise compression, without any approximations, at the scale of deep networks.
> * The layer-wise solver composed of efficient row-handling and global row merging, together with several key technical details in the implementation, such as row-wise parallelization or the handling of outliers in OBQ, is new.
> * The idea of fitting quantization into the OBS framework by progressively quantizing with exact updates.
> * Finally, we are the first to combine sparsity and quantization in the post-training setting, in particular in a mixed manner with different quantization and sparsity levels per layer. (Prior work only considered one type of compression.)
>
> Conceptually, there is already work on layer- or block-wise implementations of the Optimal Brain Surgeon. However, all these methods work via approximations, such as simply summing up individual weight updates, or ignoring the fact that the Hessian inverse actually changes with every weight deletion. Prior work argues that such approximations are **necessary** for computational tractability. By contrast, we show for the first time that an exact OBS application is not only tractable, but even quite efficient at the scale of deep networks, while yielding significant accuracy gains.
>
> > The theoretical result (Lemma 1) seems to be a direct application of the Woodbury-Sherman-Morrison formula [1] (if so, it should be cited in the text)
>
> Lemma 1 is actually not an application of the Woodbury formula, although the statements may indeed appear similar. Specifically, we do *not* perform a rank-one update here—we remove one row and column, which is not a rank-1 update. Instead, it corresponds to one iteration of Gaussian elimination, as we also stated in the Lemma statement.
>
> In fact, we initially attempted to “emulate” the same matrix transformation via a carefully crafted rank-2 update (using the rank-k version of Woodbury), but we found this to be both slower and numerically (much) less stable relative to the version we presented in the submission, which is based on this new Lemma.
>
> > I like how the paper is written, but I feel some improvements can be made to better highlight the technical contributions, and distinguish it from existing methods/ideas.
>
> We have mainly focused on describing our approach in detail, but we agree that our contributions can be better highlighted, and will attempt to do so. If the reviewer has additional specific suggestions, we would be happy to follow them.
>
> > A comprehensive comparison that includes both the setup cost (time to compress), runtime cost (inference time), and task performance (accuracy) is missing in the main text ...
>
> We did provide detailed compression time information and method comparisons in the Supplementary Material, in Appendix A.5. The main finding was that our method has comparable runtime to other post-training compression methods. There we also discuss further ways of speeding up our method’s implementation.
>
> In terms of runtime / accuracy comparisons, we agree that it would also be useful to provide baseline numbers for our final real-world mixed sparse + quantized GPU and CPU scenarios (even though we are the first to consider those particular settings). Hence, we have now added comparisons with a baseline combining the best existing independent layer-wise quantization and pruning methods (AdaPrune and AdaQuant) for all models and BOP / speedup targets. Those can be found in Section A.10 of the most recent Appendix revision or, for convenience, at this anonymized link: https://github.com/NeurIPS9406/NeurIPS9406/blob/main/comparisons.pdf. In summary, the results show that the improvements we demonstrated in the previous single-compression-type experiments also transfer to the mixed setting.
>
> If any questions remain, we are happy to engage in further discussion!

---

> > ### Comment · Reviewer_7fQD · 2022-08-08
> > **Response to the authors**
> >
> > I appreciate the authors' response, as it addressed my questions. I will raise my score from 6 to 7.

---

### Official Review · Reviewer_Xxed · 2022-07-14

**Rating:** 6
**Confidence:** 3
**Soundness:** 3 good
**Presentation:** 3 good
**Contribution:** 3 good

**Summary:**

This paper proposes a method for post-training quantization and pruning by primarily building on the Optimal Brain Surgeon (OBS) framework. It achieves this by creating an efficient version of the Hessian-based approach in OBS and applies it to quantization and pruning separately. The quantization approach is called Optimal Brain Quantizer (OBQ) and their pruning approach is ExactOBS. They evaluate their methods in multiple modes, including unstructured sparsity, block sparsity, quantization, and mixed compression. They find their method performs better or equal in most of these modes, with the added benefit in some cases of being layer-wise and more parallelizable.

**Questions:**

As correctly predicted by the authors, it is not  intuitive that one-by-one quantization of weights would give a strong benefit. How much are the (not-yet-quantized) weights being adjusted after the quantization of other weights?

There is a significant body of work on post-training quantization methods that use various amounts of calibration data, including those that use no data at all. These include HAQ, OCS, ZeroQ, QAT, etc. Are these not directly compared against because they are not solely weight quantization? Or do the baselines included claim to outperform them?

The quantized experiment mentioned using a consistent pre-trained model for comparison. Is this true with all the experiments?

**Limitations:**

No significant limitations than otherwise mentioned.

**Strengths And Weaknesses:**

=== Strengths ===

The authors include a strong motivation for their work especially from the perspective that modern hardware supports quantization and block sparsity.

It seems that in general although exact methods are usually unnecessary in machine learning, this may not be true in the context of post-training quantization and limited calibration data. This provides again a high-level motivation for pursuing an exact OBS solution.

The authors include a variety of evaluation tasks, including classification, detection, and question answering.

Paper written in an organized and clear way.

=== Weaknesses ===

In the end, what matters with compression is the accuracy vs. real-world performance curves. It would be useful to show these comparing against other methods to the extent that they are supported.

---

> ### Author Response · Authors · 2022-08-02
> **Reply to Reviewer Xxed**
>
> Thank you very much for your useful suggestions and questions, which we will now respond to in detail.
>
> > In the end, what matters with compression is the accuracy vs. real-world performance curves. It would be useful to show these comparing against other methods to the extent that they are supported.
>
> We focused on independent method-wise comparisons since the best existing methods are designed for a single compression approach (e.g. AdaPrune focuses only on pruning). Yet, we agree that it would also be interesting to provide “composite” baseline numbers for our final real-world mixed sparse + quantized GPU and CPU scenarios. To address this, we have now added results for a scheme consisting of AdaPrune for sparsification and AdaQuant for quantization (i.e. combining the best existing independent layer-wise methods) to all of our plots. Please see Section A.10 in the Appendix in the most recent revision of our Supplementary Material or, for convenience, this anonymized link: https://github.com/NeurIPS9406/NeurIPS9406/blob/main/comparisons.pdf.
>
> In summary, as expected, the individual improvements for each compression type also transfer to joint/mixed applications. This shows that our method outperforms prior approaches in terms of accuracy both for single and for composite compression.
>
> > As correctly predicted by the authors, it is not intuitive that one-by-one quantization of weights would give a strong benefit. How much are the (not-yet-quantized) weights being adjusted after the quantization of other weights?
>
> We found that, typically, about 10-15% of the weights in each layer ultimately end up with a quantized value that is different from rounding the initial weight to the nearest value. Out of those weights, only a very small fraction (at most 1-2%) are more than 1 grid-cell away from their initial round to nearest value. So updates do change weight values, but not by very large amounts on average. Additionally, we note that the early quantization of outliers, which we discussed in lines 311- 324, is also important to consistently achieve good results.
>
> > There is a significant body of work on post-training quantization methods that use various amounts of calibration data, including those that use no data at all. These include HAQ, OCS, ZeroQ, QAT, etc. Are these not directly compared against because they are not solely weight quantization? Or do the baselines included claim to outperform them?
>
> Our quantization baselines are chosen as follows: by comparison of experimental results, the best-performing post-training weight quantization method is BRECQ, while the current best sequential layer-wise method is AdaRound. Further, the current best independent layer-wise quantization method (not sequential, so in the same setup as ours) method is AdaQuant.
>
> We compared against all these methods; additionally, in Appendix A.7, we compare against BitSplit, another independent + layer-wise method, as well. All of the corresponding works show in extensive comparisons that they outperform various other existing post-training techniques, such as ZeroQ, OCS, OMSE or ACIQ. Hence, we did not consider it necessary to repeat those comparisons.
>
> As for HAQ and QAT: these are not post-training methods, since they retrain for a large number of epochs using the full dataset. Hence, a comparison with our method, that uses just 1K data samples, would not be very meaningful. Nevertheless, our method can actually be quite competitive in some cases, as we discussed this in our comparison with a recent QAT result in lines 403 - 408. For additional discussion on this post-training setting, please also see our response to Reviewer CZ4H. We will add clarifications to the next revision.
>
> > The quantized experiment mentioned using a consistent pre-trained model for comparison. Is this true with all the experiments?
>
> Yes, this is true for all our experiments. The reason we emphasize this particularly in the context of quantization is that this is unfortunately not always the case for other works in the area.
>
> If any questions remain, we are happy to engage in further discussion!

---

### Author Response · Authors · 2022-08-02
**Rebuttal Overview**

We thank the reviewers for their very useful comments. We provide detailed answers to each issue or question raised in the reviews, as review responses.

As an overview, our responses cover the following general questions raised by the reviews:

* **In response to reviewer CZ4H, we further clarify the fact that our paper focuses on post-training compression approaches.** These are defined as methods which use a very small calibration dataset (e.g., one sample per class from the training set), and perform compression in one-shot, without any retraining. This is a practical and challenging scenario, and our usage of this term is consistent with the literature.
* **This post-training scenario influences the set of methods we can fairly compare against (answering Reviewers CZ4H and Xxed).** In particular, our original submission compared against all state-of-the-art post-training compression methods, both for quantization (BRECQ, AdaRound, AdaQuant, BitSplit) and pruning (AdaPrune), relative to which we show improved or matching results. Implicitly, this means we also outperform earlier methods such as ZeroQ, OCS, OMSE or ACIQ. We did not compare against retraining methods, as they use orders of magnitude more data, and often perform gradual pruning. This is standard across the post-training literature.
* **Nevertheless, to address questions by reviewer CZ4H, we demonstrate via preliminary experiments that our method can be favorably combined with retraining as well.** In particular, we show that OBC pruned models outperform Movement Pruning on the BERT/SQuAD task when they are finetuned for the same number of epochs, even though we perform pruning in one-shot, rather than gradually for MvP.
* **Further, we have added side-by-side comparisons against a combination of state-of-the-art methods for all of our mixed real-world sparse + quantized GPU and CPU scenarios, addressing suggestions by Reviewers 7fQD and Xxed.** These results can be found in Section A.10 of the Appendix in the revised Supplementary Material (or, for convenience, at this anonymized link: https://github.com/NeurIPS9406/NeurIPS9406/blob/main/comparisons.pdf ) and show that, as expected, the improvements observed in our previous single compression-type experiments transfer to the combined setting as well.
* **Finally, we provide a thorough discussion of the additional references provided by Reviewer CZ4H.** We will cite these works in the next paper revision, although we emphasize that none of these references affects our initial claims.

We would be very happy to engage in further discussion.

With best regards,
The authors

---

### Author Response · Authors · 2022-08-07
**Rebuttal Follow-up**

Dear Reviewers and ACs,

As the discussion period is quickly drawing to an end, we would like to gently follow-up on our rebuttal.

We believe our responses do address the questions raised in a substantive manner (e.g. in the form of pointers to results the reviewers may have missed, additional experimental results requested by reviewers, and detailed clarifications), and we would welcome the opportunity to discuss these or any additional questions the reviewers may have.

With best regards,

The authors

---

### Meta-Review · Area_Chair_VU4c · 2022-08-28

**Recommendation:** Accept
**Confidence:** Certain

**Metareview:**

The reviewers have reached a consensus in favor of accepting this paper, and I agree with this consensus. This is a technically solid paper that makes a good contribution to the field of post-training compression. The issues brought up in the reviews were adequately addressed by the author response, and I expect the final version of the paper will clarify some of the confusion regarding comparison with retraining.

**Award:**

No

---

### Decision · Program_Chairs · 2022-09-14

Accept